# Cognitive and Cellular Effects of Combined Organophosphate Toxicity and Mild Traumatic Brain Injury

**DOI:** 10.3390/biomedicines11051481

**Published:** 2023-05-19

**Authors:** Dor Freidin, Meirav Har-Even, Vardit Rubovitch, Kathleen E. Murray, Nicola Maggio, Efrat Shavit-Stein, Lee Keidan, Bruce A. Citron, Chaim G. Pick

**Affiliations:** 1Department of Anatomy and Anthropology, Sackler School of Medicine, Tel Aviv University, Tel Aviv 6997801, Israel; dofreidin@gmail.com (D.F.); meiravhe@mail.tau.ac.il (M.H.-E.);; 2Laboratory of Molecular Biology, VA New Jersey Health Care System, Research & Development, East Orange, NJ 07018, USA; 3Rutgers School of Graduate Studies, Newark, NJ 07103, USA; 4Department of Neurology, The Chaim Sheba Medical Center, Ramat Gan 52626202, Israel; 5Department of Neurology and Neurosurgery, Sackler Faculty of Medicine, Tel Aviv University, Tel Aviv 6997801, Israel; 6Sagol School of Neuroscience, Tel Aviv University, Tel Aviv 6997801, Israel; 7Department of Pharmacology, Physiology & Neuroscience, Rutgers New Jersey Medical School, Newark, NJ 07103, USA; 8Sylvan Adams Sports Institute, Tel Aviv University, Tel Aviv 6997801, Israel; 9The Dr. Miriam and Sheldon G. Adelson Chair and Center for the Biology of Addictive Diseases, Tel Aviv University, Tel Aviv 6997801, Israel

**Keywords:** mTBI, organophosphates, cognitive and behavioral tests, neuronal loss, neuroinflammation

## Abstract

Traumatic brain injury (TBI) is considered the most common neurological disorder among people under the age of 50. In modern combat zones, a combination of TBI and organophosphates (OP) can cause both fatal and long-term effects on the brain. We utilized a mouse closed-head TBI model induced by a weight drop device, along with OP exposure to paraoxon. Spatial and visual memory as well as neuron loss and reactive astrocytosis were measured 30 days after exposure to mild TBI (mTBI) and/or paraoxon. Molecular and cellular changes were assessed in the temporal cortex and hippocampus. Cognitive and behavioral deficits were most pronounced in animals that received a combination of paraoxon exposure and mTBI, suggesting an additive effect of the insults. Neuron survival was reduced in proximity to the injury site after exposure to paraoxon with or without mTBI, whereas in the dentate gyrus hilus, cell survival was only reduced in mice exposed to paraoxon prior to sustaining a mTBI. Neuroinflammation was increased in the dentate gyrus in all groups exposed to mTBI and/or to paraoxon. Astrocyte morphology was significantly changed in mice exposed to paraoxon prior to sustaining an mTBI. These results provide further support for assumptions concerning the effects of OP exposure following the Gulf War. This study reveals additional insights into the potentially additive effects of OP exposure and mTBI, which may result in more severe brain damage on the modern battlefield.

## 1. Introduction

Traumatic brain injury (TBI) occurs when the head is ompacted by an object or external force. TBIs are primarily caused by road accidents, falls, wars, assaults, and sports injuries [1]. Several mechanisms of damage may cause TBIs, including blunt injury (when there is significant acceleration or malformation of the brain tissue), penetrating injury (when the invasion of the skull causes damage), blast injury, and concussion [2]. The pathophysiology of TBI can be divided into primary and secondary injuries. Primary injury occurs when an outside force is applied to the brain, directly affecting neural tissue, glial cells, and the vascular system based on their physical characteristics. Secondary injuries result from the progression of biological events triggered by the primary injury. They can include ischemia, glutamate toxicity, neuroinflammation, edema, increased permeability of the blood–brain barrier, oxidative stress, and cellular dysfunction leading to apoptosis [3].

Currently, there is a dearth of effective interventions designed to address the secondary injury cascades that occur following an initial traumatic event. However, prior research has suggested that a multifaceted approach incorporating a combination of therapeutic modalities, supplementation strategies, and pharmaceutical agents may yield promising results in mitigating the effects of these secondary injuries [4].

Head injuries are commonly categorized by the severity of injury according to the Glasgow Coma Scale (GCS), which assesses a patient’s condition based on their eye, motor, and verbal responses. This test distinguishes mild, moderate, and severe injuries [5].

Severe TBI can be diagnosed relatively easily by characterizing brain tissue damage. In contrast, blood–brain barrier disruption and the development of edema following a mild traumatic brain injury (mTBI) are more challenging to assess because routine tests, including imaging, fail to show changes in brain structure [6]. Additionally, many patients do not lose consciousness after the injury [7]. More than 80% of head injury cases are classified as mild TBIs. In most cases of mild neurotrauma, immediate symptoms gradually disappear within a year following the trauma. However, patients sometimes suffer from persistent and long-lasting neurocognitive impairments, including various cognitive, emotional, and behavioral disorders [8]. mTBI is a common injury in active combat zones and areas subject to frequent terrorist attacks, typically caused by proximity to breaching devices, heavy weaponry, and improvised explosive devices (IEDs), affecting both civilians and military personnel [9,10].

Organophosphates (OPs) are a group of toxic, broad-impacting chemicals with various uses ranging from chemical warfare agents (e.g., Soman, Sarin, Tabun, Cyclosarin, VX) to pesticides. The human body can absorb OPs through inhalation, digestion, or cutaneous penetration. Exposure to OP-based pesticides accounts for 3 million cases per year, rendering insecticide poisonings common in developing countries. Previous studies have demonstrated that acute poisoning from exposure to OP-based pesticides can cause adverse health effects, including vasomotor and verbal memory deficits [11,12]. The effects of OP poisoning are dependent on several factors, including the specific type of OP, the amount of OP to which an individual is exposed, the route of exposure, the duration of exposure, and the age of the individual [13].

OP poisoning poses a life-threatening primary effect, primarily affecting the peripheral nervous system. OP inhibits the enzyme acetylcholinesterase (AChE) in an irreversible manner, which can lead to damage to the cholinergic system. Immediate clinical signs of OP poisoning include tremors, paralysis, and even death due to respiratory failure. Furthermore, OP poisoning may also cause immediate and long-term damage to the central nervous system [14]. Within the brain, the cholinergic system is essential for learning, memory, and consciousness [15,16,17]. Consequently, exposure to OPs may impair memory, consciousness, and motor and emotional abilities [18,19,20].

In recent years, conventional injuries (e.g., explosives) and unconventional exposures, such as chemical weapons (e.g., civil war in Syria), have occurred more frequently in combat zones and terrorist-stricken areas. As such, the likelihood that civilians, military, and law enforcement agents are exposed to either insult or experience a combined exposure to both OPs and mTBI has significantly increased. This combined insult makes it challenging to determine the etiology of the injury and to provide appropriate treatment. In addition, there is currently no research-based medical–therapeutic protocol to treat this combined injury.

In the present study, we utilized an animal model to quantify the extent of damage from co-exposure to OP poisoning and mTBI and examined their combined effects on cerebral functioning over time. These effects were assessed using a series of accepted behavioral and immunohistochemical analyses.

## 2. Materials and Methods

### 2.1. Animals

Male ICR mice at 6–8 weeks of age (30–40 g) were purchased from Invigo RMS Inc. (Ein Karem Jerusalem, Israel). The mice were kept at room temperature (22 ± 1℃) on a 12-h light/dark cycle with five mice per cage (32 × 21.5 × 12 cm^3^). Access to standard rodent chow (Purina, Neenah, WI, USA) and water was unrestricted (ad libitum). All experimental procedures were conducted during the light phase. The cage bedding was sawdust replaced twice a week simultaneously for all the cages. Upon arrival at the veterinary service center, the animals were provided with three days of recovery and acclimatization to the new location. Two days prior to the experiment, all cages were moved to the experimental room for habituation and anxiety reduction. The Ethics Committee of the Sackler Faculty of Medicine approved the experimental protocol (01–16-058) in compliance with the guidelines for animal experimentation of the National Institutes of Health (DHEW publication 85–23, revised, 1995).

### 2.2. Experimental Groups

The study consisted of five experimental groups. Each group included 10–12 mice for the behavioral and cognitive tests and 5 mice for the immunohistochemistry analysis (total of *n* = 85). To avoid behavioral confounds, each group of animals was tested once. Previously, a variance analysis was conducted to determine how many animals were required in each assessment group and how long the measurements lasted. Several previous studies related to the subject were considered when determining sample size [21,22].

The animals were divided into TBI experimental groups receiving either mTBI or sham insults, as well as paraoxon groups receiving either paraoxon or a vehicle. The experimental design was as follows:A.Control group (sham, vehicle): The treatment conditions were identical, including anesthetization by inhalation of isoflurane vapor for several minutes and an intraperitoneal (IP) injection of 1 mL of saline solution.B.mTBI group (TBI, vehicle): The treatment conditions were identical. The mice were exposed to mild traumatic brain injury and IP injection of 1 mL of saline solution.C.Paraoxon group (sham, paraoxon): The treatment conditions were identical, including anesthetization by inhalation of isoflurane vapor for several minutes. Paraoxon was diluted in 0.9% saline and absolute ethanol (dehydrated, 99%) and administered by IP injection at a dose of 0.3 mg/kg.D.mTBI + paraoxon group (TBI, post-paraoxon): The mice were exposed to mTBI followed by paraoxon administration after 1 h. Paraoxon was diluted in 0.9% saline and absolute ethanol (dehydrated, 99%) and administered by IP injection at a dose of 0.3 mg/kg.E.Paraoxon + mTBI (TBI, pre-paraoxon): The mice were exposed to paraoxon followed by mTBI after 1 h. Paraoxon was diluted in 0.9% saline and absolute ethanol (dehydrated, 99%) and administered by IP injection at a dose of 0.3 mg/kg.

### 2.3. Mouse Closed-Head Mild Traumatic Brain Injury

Mild traumatic brain injury (mTBI) was implemented in accordance with past protocols conducted by our group [21]. The head injury was induced by a concussive head trauma device, which involves a fixed weight freefalling along a defined trajectory. The device consisted of a hollow aluminum tube (80 cm in length and 13 mm in diameter). The weight (10 g) and height from which the metal weight was dropped determined the severity of the injury. At the time of injury, mice were placed on a spongy surface with the tube vertically above their heads; this allowed the head to move parallel to the plane of injury during the weight drop, thus simulating a head injury condition. Deliberate trauma was caused specifically to the fronto-lateral area on the right side of the head (midway between the ear and the right eye) [23]. This model was chosen to simulates diffuse traumatic brain injury, which is characteristic of road accidents or falls. After the injury, the mice were assessed using the Neurological Severity Score (NSS) scale to confirm the absence of any severe acute neurological injuries [21].

### 2.4. Paraoxon Administration

Mice received IP injections of a single dose of paraoxon (N-12816, Sigma-Aldrich, Rehovot, Israel, 0.3 mg/kg). The paraoxon dilution was performed in a chemical hood. Paraoxon was first diluted with propylene glycol to 50 mg/mL, and then further diluted with saline to 1.36 mg/mL. Paraoxon was chosen as a representative substance for the organophosphate group due to our familiarity with this substance through previous work, and because paraoxon is easy to handle and administer with low collateral damage. The dosage given to the animals followed the protocol described by Golderman et al., who treated the animals at a dose of 0.5 mg/kg. Our protocol reduced the dosage to 0.3 mg/kg to ensure that seizures would not be induced in combination with the onset of mTBI [24].

### 2.5. Behavioral and Cognitive Tests

All behavioral and cognitive tests were performed in succession 30 days post-injury. Figure 1).

The test arenas used for the EPM, NOR, and Y-maze were manufactured to meet the specific size requirements of our group and have been extensively validated throughout our previous studies. The time spent in each part of the arena was manually measured by a double-blinded researcher.

#### 2.5.1. Elevated Plus Maze (EPM)

The EPM test is used to estimate anxiety behavior in rodents. The test capitalizes on the conflict between the innate fear of rodents in open spaces and their curiosity and desire to explore a new environment [25,26]. The maze consisted of two open arms (30 × 5 × 0.25 cm^3^) and two closed arms (30 × 5 × 15 cm^3^) made of Polymethyl Methacrylate (PMMA). Each pair of arms faced each other on a 50 cm-high surface, forming a “+” shape. At the beginning of the test, we placed the animal in the center of the platform facing one of the open arms and allowed the animal to explore the maze for 5 min. This test measured the time the animal spent in the open arms and the number of times it entered the open space.

#### 2.5.2. Novel Object Recognition (NOR) Test

The NOR test is based on the natural curiosity of rodents to explore new objects and is intended to test the visual memory of the animals [27]. The arena was a square surface (60 × 60 cm) with high walls (20 cm). The NOR test consisted of three steps, with a 24-h interval between each step: (A) Acclimatization step: The tested mouse was placed into the arena for five minutes to become acclimated to the arena itself. (B) Learning step: The mouse was placed into the arena with two identical “old” objects for five minutes to familiarize the mouse with the objects. (C) Test step: The mouse was placed into the arena with one “old” object from the learning phase and one “new” object for five minutes. Between each animal, the surface and objects were cleaned with ethanol to minimize the odors left by the previous animals. The Aggelton index [23] was calculated to assess the degree of learning and visual memory of the animals according to the following formula:Time exploring new object − time exploring old object/time exploring new object + time exploring old object = preference index.(1)
(2)$$\frac{time\;A\;object-time\;B\;object}{time\;A\;object+time\;B\;object}=Preference\;index$$

A higher preference index indicated better recall. Animals that explored the objects for less than 10% of the total time spent in the arena (i.e., less than 30 s with the two objects together) were excluded from the statistical calculations because it is not possible to estimate the visual memory level of a mouse that does not engage with the objects at all.

#### 2.5.3. Y-Maze

The Y-maze test is used to evaluate short-term spatial memory and relies on the animal’s preference for exploring a new place [23]. The maze was made of black Perspex and had 3 arms (8 × 15 × 30 cm) arranged at a 120° angle. One arm was randomly selected as the “start arm”. First, each animal was placed on the outer edge of the “start arm” with one of the remaining two arms blocked. The blocked arm was defined as the “new arm”, while the accessible arm was defined as the “old arm”. The animal was given five minutes to freely explore the two open arms. At the end of the allotted time, the animal was returned to its cage for two minutes. During this time, the maze was cleaned with ethanol to remove any traces left by the animal. After two minutes, the animal was returned to the maze and allowed to freely roam in all three arms for two additional minutes. The time that the animal spent in each arm was measured during these two minutes to assess the animal’s ability to distinguish between the “new” and the “old” arms. To avoid any bias due to individual preferences for a specific arm, we changed the place of the “new” arm between animals. During the test phase of the experiment, naïve (untreated) animals were expected to prefer the “new” arm over the “old” arm due to their natural curiosity to explore a new area. The ability to distinguish between the “new” and “old” arms depends on the spatial memory of the animal. The Aggleton index [28] was calculated to assess spatial memory according to the following formula:(3)$$\frac{time\;in\;new\;arm-time\;in\;old\;arm}{time\;in\;new\;arm+time\;in\;old\;arm}=Preference\;index$$

An animal with intact spatial memory will display a high preference index, whereas an animal with impaired spatial memory will have a low preference index.

### 2.6. Immunohistochemistry

Immunohistochemical studies were performed on hippocampal (dentate gyrus hilus—DGH) and temporal cortex (Cx) tissue sections obtained from animals euthanized 30 days post-injury. The mice were anesthetized with ketamine (100 mg/kg) and xylazine (10 mg/kg) and underwent transcardiac perfusion with 10 mL of phosphate-buffered saline (PBS) followed by 20 mL of 4% paraformaldehyde (PFA) in 0.1 M PBS at pH 7.4. The brains were removed, fixed overnight in 4% PFA, and then placed in 1% PFA. The brains were prepared in a multiblock orientation by Neuroscience Associates (Knoxville, TN, USA), and 35 μm sections were collected successively through the brains. Floating section staining and mounting were performed using the antibodies detailed in Table 1. Microscopy was performed using a Fluoview 3000 laser scanning confocal microscope (Olympus, Waltham, MA, USA). The marked locations were determined on a stitched map with only Hoechst 33342 (ThermoFisher, Asheville, NC, USA) staining captured. For all analyses, regions of interest were selected on the stitched map, which depicted only the nuclear staining and was blinded with respect to groupings. These regions were then collected by multi-area routines and sequenced with Fluoview 3000 version 2.5.1 software (Olympus, Waltham, MA, USA) without intervention. The images were created by combining as Z stacks and maximum Z projections of coronal sections centered around approximately −2.9 mm from Bregma. These sections were identically illuminated (405, 488, 561, and 640 nm diode lasers) and detected. Automated analysis of cell morphology, intensities, and cells counts was conducted using cellSens version 18.0 (Olympus, Waltham, MA, USA) and ImageJ version 1.52a (NIH, Bethesda, MD, USA) [29] software. Macros were employed to automate image analyses. Autothresholding was used to prevent bias using the Li Dark algorithm for NeuN counting and the RenyiEntropy algorithm for astrocyte morphology. Counting was accomplished by converting NeuN images to binary and running the Watershed tool to separate any cells in contact, then executing the Analyze Particles tool. For cell morphology analysis, skeletons were produced with the Skelotonize tool after thresholding, and process characteristics were determined using the Analyze Skeleton (2D/3D) tool.

### 2.7. Statistics

All values are presented as mean values ± standard deviation. Statistical calculations were performed using IBM SPSS version 24 (Genius Systems, Petah Tikva, Israel). The behavioral data were analyzed using ANOVA tests for continuous variables. For more detailed data, LSD post hoc tests were used. Statistically significant differences between the averages were indicated as * *p* < 0.05, ** *p* < 0.01, *** *p* < 0.001.

## 3. Results

### 3.1. Behavioral Effects of Closed-Head mTBI Caused by 10-g Weight Drop

#### 3.1.1. Anxiety Measured with the Elevated Plus Maze (EPM)

The EPM test was applied to assess anxiety-like behavior. A one-way ANOVA demonstrated no significant main effect of group [F(4, 50) = 2.008, NS = 0.108].

Abnormal anxiety behavior was ruled out in all groups. mTBI exposure did not affect anxiety.

#### 3.1.2. Recognition Memory Evaluated by Novel Object Recognition (NOR)

The NOR test was used to assess visual recognition memory (Figure 2A). A one-way ANOVA using LSD post-hoc analysis [F(4, 50) = 9.886, *p* = 0.000] did not show a significant difference between the control and mTBI groups (*p* = 0.620), indicating that the mTBI induced by the 10-g (“light”) weight had a negligible effect when compared with the uninjured mice. The control group performed significantly better on this task than the paraoxon (*p* = 0.002), paraoxon + mTBI (*p* < 0.001), and mTBI + paraoxon groups (*p* < 0.001). We found a significant difference between the mTBI and the mTBI + paraoxon (*p* < 0.001) groups, as well as the mTBI and paraoxon + mTBI (*p* = 0.003) groups; however, no difference was observed between mTBI and paraoxon alone. In addition, we found that there was a significant difference between the paraoxon and mTBI + paraoxon groups (*p* = 0.042). There were no significant differences in preference index between the paraoxon + mTBI and the mTBI + paraoxon groups (*p* = 0.887).

#### 3.1.3. Spatial Memory Tested with the Y-Maze

The Y-maze test was used to assess spatial memory (Figure 2B). A one-way ANOVA using LSD post hoc analysis [F(4, 50) = 6.729, *p* = 0.000] showed that mice in the mTBI + paraoxon (*p* < 0.001) and paraoxon + mTBI (*p* < 0.001) groups performed significantly worse than mice in the control group.

Similar to our findings with the NOR test, we also found that the mTBI group performed significantly better than the mTBI + paraoxon group (*p* = 0.007). We found that the paraoxon group performed significantly worse during Y-maze testing when compared to the control group (*p* = 0.018). Moreover, a significant difference was found between the paraoxon and mTBI + paraoxon groups (*p* = 0.027). No significant difference was found between either the control and mTBI groups or between the mTBI and paraoxon groups.

### 3.2. Combined Insult with mTBI and Paraoxon Induces Neuronal Loss

The number of NeuN^+^ neurons in the temporal cortex (Figure 3A) was analyzed using a one-way ANOVA with an LSD post hoc test: F(4, 20) = 5.618, *p* = 0.003]. The total number of NeuN^+^ neurons within the temporal cortex was significantly lower in the paraoxon + mTBI and mTBI + paraoxon groups compared to the control group (*p* = 0.008, *p* = 0.028). The mTBI mice had significantly more neurons than paraoxon + mTBI and mTBI + paraoxon (*p* = 0.001, *p* = 0.003). The combined groups of paraoxon + mTBI and mTBI + paraoxon showed significantly more neuronal damage than paraoxon alone (*p* = 0.009, *p* = 0.03).

The number of NeuN^+^ neurons in the dentate gyrus (Figure 3B) was analyzed using a one-way ANOVA with an LSD post hoc analysis: [F(4, 20) = 1.945, NS = 0.142]. Significant differences were found in the DGH between the mTBI and paraoxon + mTBI groups (*p* = 0.049). No significant difference was found between mTBI and mTBI + paraoxon groups.

Images of immunohistochemistry staining in the temporal cortex and DGH with NeuN^+^ cells shown in red are presented in Figure 3C.

### 3.3. Paraoxon Elevates Neuroinflammatory Responses, as Indicated by Reactive Astrocytosis

Astrocyte reactivity was examined in the dentate gyrus (Figure 4B). A one-way ANOVA with an LSD post-hoc test was used to analyze GFAP intensity [F(4, 20) = 3.438, *p* = 0.027], GFAP^+^ astrocyte counts [F(4, 20) = 2.755, *p* = 0.056], and astrocyte morphology [F(4, 20) = 3.294, *p* = 0.032]. The results indicated that the GFAP intensity in the DGH was higher in the mTBI, paraoxon, paraoxon + mTBI, and mTBI + paraoxon groups than in the control group (*p* = 0.013, *p* = 0.002, *p* = 0.038, and *p* = 0.027, respectively).

GFAP counts in the DGH (Figure 4D) revealed significant changes in astrocyte counts in the dentate gyrus between the paraoxon and control groups compared with mTBI animals (*p* = 0.018, *p* = 0.019, respectively). Astrocyte morphology in the DGH (Figure 4F) showed marked alterations in astrocyte morphology in the dentate gyrus of paraoxon + mTBI mice vs. controls (*p* = 0.003). Astrocyte morphology was also significantly different between the mTBI and paraoxon + mTBI groups, and between the paraoxon and paraoxon + mTBI groups (*p* = 0.019, *p* = 0.017, respectively).

Images of immunohistochemistry staining in the temporal cortex and DGH with GFAP^+^ cells shown in green are presented in Figure 4G. No statistically significant differences in neuroinflammatory responses were found in the temporal cortex as measured by GFAP intensity [F(4, 20) = 1.001, NS = 0.430], counts [F(4, 20) = 1.288, NS = 0.308], or morphology [F(4, 20) = 1.319, NS = 0.297] (Figure 4A,C,E).

## 4. Discussion

Recently, we have witnessed an increase in military conflicts. The resurgence of countries on the battlefield and the hazards related to the use of chemical or biological weapons highlight the importance of understanding the combined damage caused by OPs and mTBI. In this study, we traced patterns of damage that characterize these combined injuries. Using our mouse model, we tested spatial and visual recognition memory (NOR and Y-maze behavioral tests, Figure 2) at 30 days post-injury. We found that mTBI caused by a relatively minor weight drop (10 g) alone did not have any observable effect compared with the sham injury. In contrast, all groups exposed to paraoxon in combination with mTBI exhibited cumulative damage.

These results provide several valuable insights. First, the use of a 10-g weight to induce mTBI resulted in minimal to non-existent damage, confirming that the trauma given was indeed mild, as previously demonstrated by Tashlykov et al. (2009) [6]. This finding aligns with results from prior experiments in our laboratory which utilized weights up to 70 g [22,30,31]. Previous protocols that included weights from 30–70 g induced significant cognitive damage in the groups that received only the mTBI intervention compared to the control groups. In this study, no significant cognitive dysfunction was observed following a 10-g weight drop. However, mice exposed to paraoxon demonstrated significant spatial and visual memory impairments. These results indicate that isolated paraoxon exposure had a significant effect on memory, and that combined mTBI and paraoxon exposure resulted in additive repercussions. This was reflected both in the results of the behavioral tests and according to the immunohistochemistry tests in the cortex using NeuN antibody and DGH. In addition, immunohistochemical analysis (Figure 3 and Figure 4) revealed that paraoxon exposure induced neuronal loss in the temporal cortex and neuroinflammatory reactions in the hippocampus at 30 days post-injury. These results are supported by literature showing that paraoxon exposure in a rat model reduced the survival rate of neurons and astrocytes in the cortex [32]. There was a modest reduction in NeuN+ cell density in the cortex after paraoxon + mTBI and after mTBI + paraoxon exposure relative to the control, while a similar trend was only found in the dentate gyrus hilus. Further research is warranted to investigate the underlying mechanisms responsible for the responses that we observed, such as whether the loss of neurons and astrocytes involves apoptosis or other mechanisms.

Research conducted after the Gulf War, during which Sarin (OP) or other toxicant exposures were likely combined with mTBI, supports our findings [33,34,35]. Approximately 25–32% of U.S. Veterans of the Gulf War suffered from a disorder known as Gulf War illness (GWI), which is characterized by multiple symptoms including fatigue, headaches, cognitive impairment, and musculoskeletal pain [33]. These studies examined the combined effect of mTBI + CBW (chemical–biological weapon) and demonstrated that such damage is associated with chronic morbidity, similar to the etiology of OP-induced damage combined with mTBI. Our observations significantly reinforce these findings by confirming tissue damage, which was not possible in early studies based on imaging and clinical evaluations of Veterans. The hypothesis that emerged from early studies was that GWI might be a disorder that is also expressed neuropathologically. More recently, animal studies have indicated that the effects of toxicant exposure in combination with TBI can lead to more severe consequences in the brain [36]. Both our behavioral and immunohistochemistry results support the additive effect of the combined insults. The significance of such a combined effect is important, given that the battlefield may expose individuals to unusual combinations of insults.

This study found no difference between paraoxon exposure before or after mTBI, but pronounced differences were observed when comparing both groups to the controls. This finding may support previous studies suggesting isoflurane administration as a treatment for neurotoxicity, as isoflurane was administered in proximity to paraoxon exposure as anesthesia prior to injury, potentially reducing brain damage [37,38]. However, this effect was not supported by immunohistochemistry, as we observed greater neuronal damage after paraoxon exposure than after TBI, and more neuroinflammation was present when paraoxon exposure occurred after TBI.

The hippocampus (DGH) was selected for immunohistochemical analysis due to its role in memory and learning to understand and further investigate the deficits observed in the cognitive tests. Additionally, the temporal cortex was examined as the region of impact.

Spatial and visual memory deficits were more pronounced when paraoxon was administered after mTBI compared with exposure to mTBI alone. This finding supports the existing hypothesis that mTBI can cause blood–brain barrier dysfunction. Therefore, the combination of mTBI and paraoxon is more detrimental to spatial and visual memory [39].

Analysis of GFAP intensity in astrocytes showed that mTBI caused diffuse damage to the DGH, but no mechanical damage was found in the temporal cortex. In particular, in the dentate gyrus region, neuroinflammatory responses may account for the paraoxon-induced loss of neurons while GFAP immunoreactivity was elevated. Astrocyte numbers were similar between controls and paraoxon-treated animals. As shown in Figure 4B, paraoxon exposure alone resulted in greater damage than control, paraoxon + mTBI, or mTBI + paraoxon insults. The effects of paraoxon may have been mitigated by exposure to isoflurane, but this result is somewhat unclear.

In summary, these results suggest that paraoxon exposure, particularly when combined with mild TBI, may significantly affect neurological functions, including both spatial and visual memory. Animals that received both paraoxon and mTBI insults exhibited the most significant cognitive and behavioral deficits, suggesting an additive effect. Mice exposed to paraoxon with or without mTBI experienced a decrease in cortical neuronal survival, while those exposed to paraoxon and then mTBI exhibited a decrease only in the dentate gyrus hilus. All groups exposed to mTBI and/or paraoxon demonstrated increased neuroinflammation in the dentate gyrus. Animals exposed to paraoxon followed by mTBI showed significant changes in astrocyte morphology. Additional inflammatory markers, e.g., CD68 and gene expression changes, could be examined in future studies. These findings validate assumptions about the effects of paraoxon exposure in both war settings and pesticide exposure, while shedding light on the potentially more severe, additive damage resulting from a combination of OP and mTBI in modern warfare and exposure to pesticides.

## Figures and Tables

**Figure 1 biomedicines-11-01481-f001:**
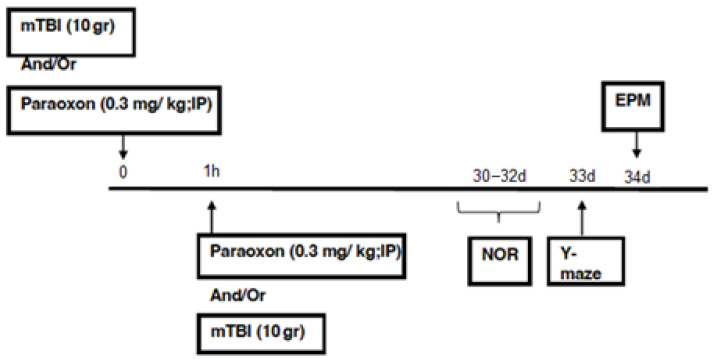
Timeline.

**Figure 2 biomedicines-11-01481-f002:**
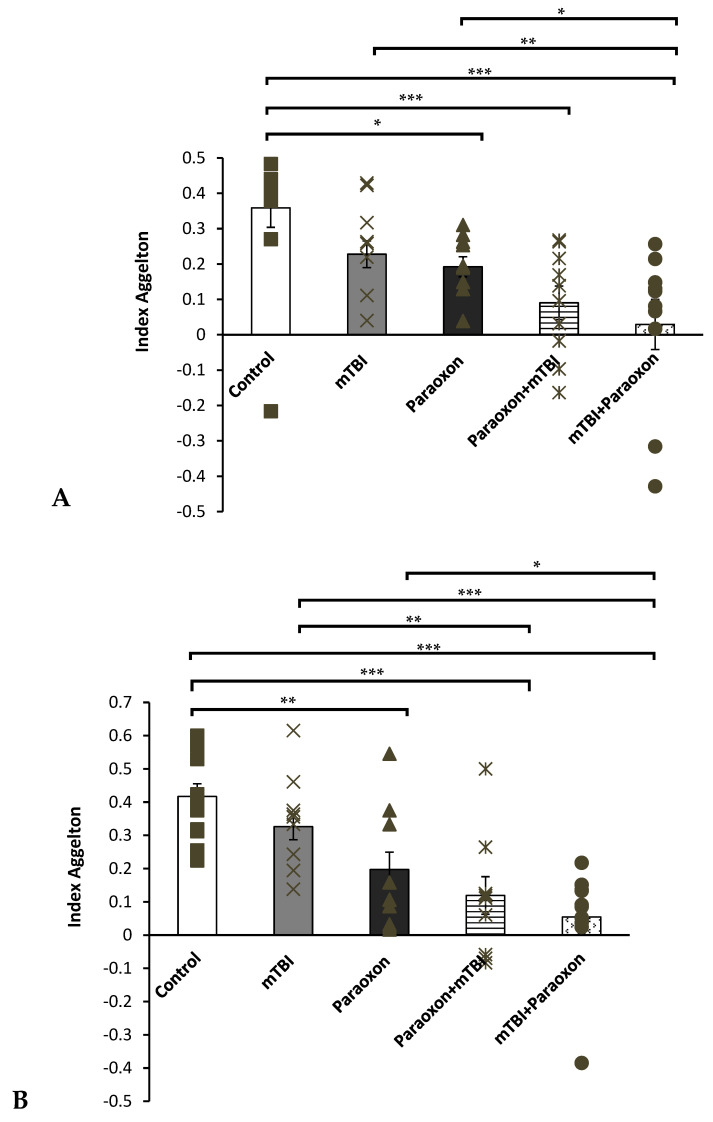
Behavioral test scores. (**A**) NOR test: differences in visual recognition memory performance between mice in the control (*n* = 12), mTBI (*n* = 12), paraoxon (*n* = 11), paraoxon + mTBI (*n* = 10), and mTBI + paraoxon (*n* = 10) groups. (**B**) Y-maze test: differences in spatial memory performance between mice in the control (*n* = 12), mTBI (*n* = 12), paraoxon (*n* = 11), paraoxon + mTBI (*n* = 10), and mTBI + paraoxon (*n* = 10) groups. * *p* < 0.05, ** *p* < 0.01, *** *p* < 0.001.

**Figure 3 biomedicines-11-01481-f003:**
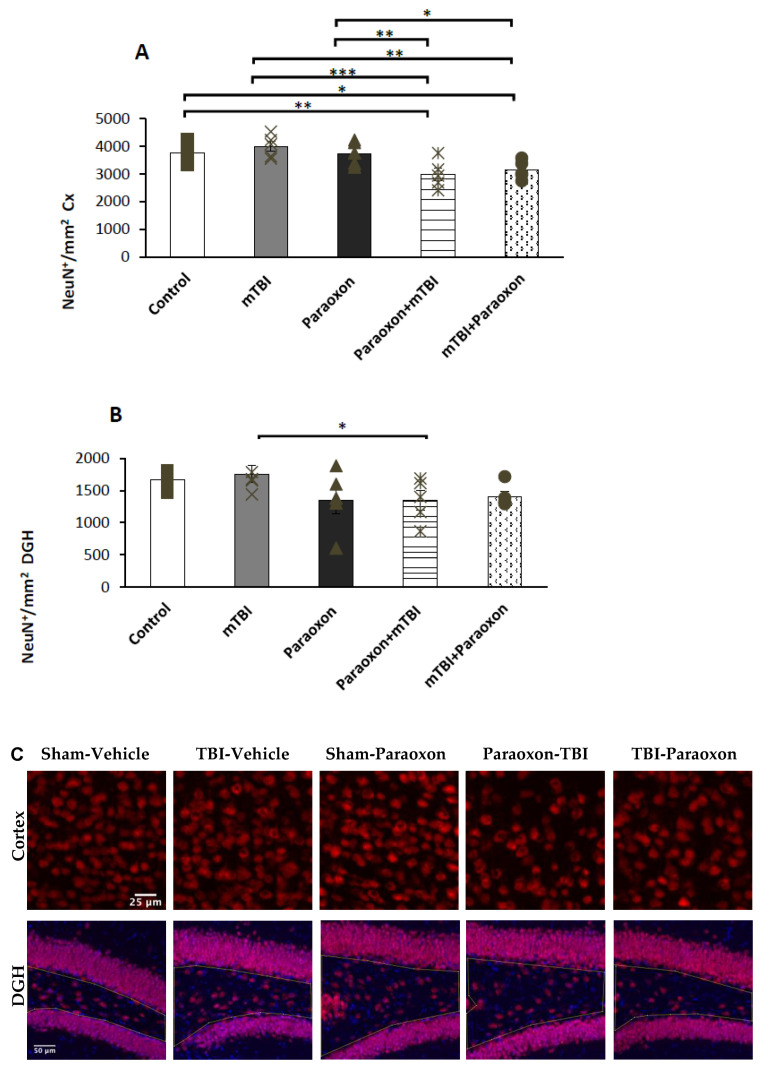
NeuN^+^ counts in the cortex and hippocampus of control (*n* = 5), mTBI (*n* = 5), paraoxon (*n* = 5), paraoxon + mTBI (*n* = 5), and mTBI + paraoxon (*n* = 5) mice. Paraoxon exposure before or after mTBI led to a significant decrease in the density of NeuN^+^ neurons compared to control and mTBI tissues in the cortex. (**A**) Quantification of total surface area labeled with NeuN in the temporal cortex. (**B**) Quantification of total surface area labeled with NeuN in the DGH. (**C**) Representative images of immunohistochemical staining in the temporal cortex (upper panel) and DGH (lower panel). NeuN+ cells are shown in red, and nuclei are shown in blue. Yellow lines outline the hilus region of the DG. * *p* < 0.05, ** *p* < 0.01, *** *p* < 0.001.

**Figure 4 biomedicines-11-01481-f004:**
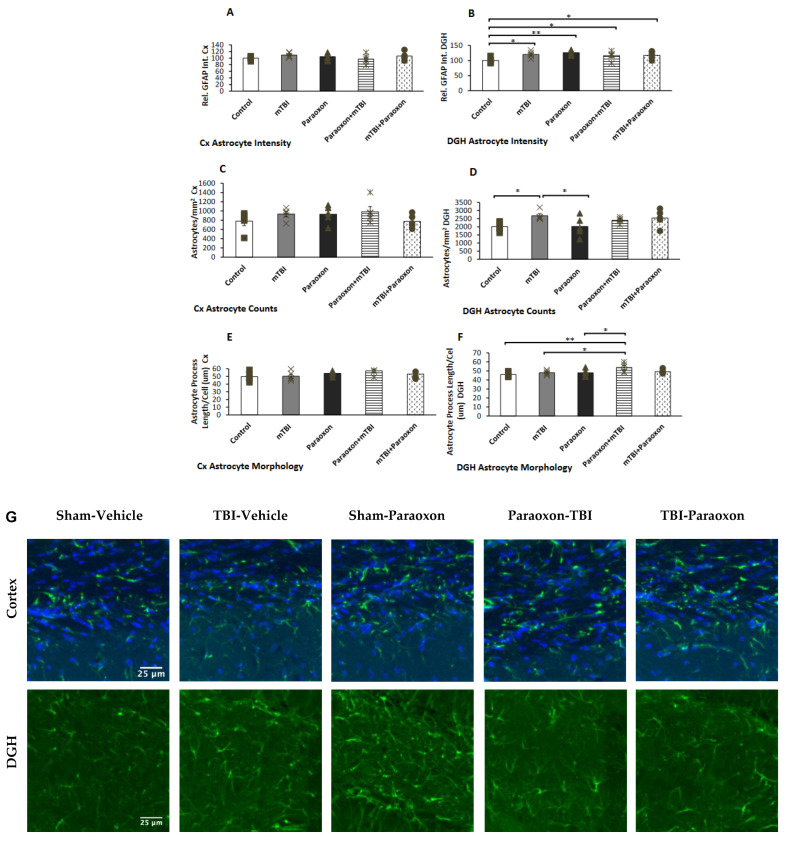
Astrocyte changes in the cortex and hippocampus of control (*n* = 5), mTBI (*n* = 5), paraoxon (*n* = 5), paraoxon + mTBI (*n* = 5), and mTBI + paraoxon (*n* = 5) mice. Paraoxon increases active astrocyte expression in the DGH only and changes astrocyte morphology before mTBI. The graphs present quantifications of: (**A**) GFAP intensity in the temporal cortex; (**B**) GFAP intensity in the DGH; (**C**) astrocyte counts in the temporal cortex; (**D**) astrocyte counts in the DGH; (**E**) astrocyte morphology in the temporal cortex; and (**F**) astrocyte morphology in the DG. Representative images of immunohistochemical staining in the DGH and the temporal cortex are presented in (**G**). GFAP-positive cells are shown in green, and nuclei are shown in blue. Scale bars are 25 μm. * *p* < 0.05, ** *p* < 0.01.

**Table 1 biomedicines-11-01481-t001:** Immunohistochemistry reagents.

Target/Fluorochrome	Primary/Secondary	Probe	Manufacturer	Catalog	Dilution
Nuclei		Hoechst 33342	ThermoFisher (Asheville, NC, USA)		
Astrocytes	Primary	Chicken anti-GFAP	Encor (Gainesville, FL, USA)	CPCA-GFA	1:1500
Alexa488	Secondary	Donkey anti-chicken	Jackson (West Grove, PA, USA)	703–545-155	1:500
NeuN + Neurons	Primary	Rabbit anti-NeuN	Abcam (Cambridge, MA, USA)	Ab104225	1:5000
Alexa647	Secondary	Donkey anti-rabbit	ThermoFisher (Asheville, NC, USA)	A31573	1:500

## Data Availability

Not applicable.

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
