# Peer review of "Cognitive and Cellular Effects of Combined Organophosphate Toxicity and Mild Traumatic Brain Injury"

_biomedicines, 2023, doi:10.3390/biomedicines11051481_

Round 1

Reviewer 1 Report

In this work, the authors conduct an interesting investigation of the synergic effect of biological weapons and mild traumatic brain injury. Even though this is a very descriptive article with not many explorations of biological mechanisms, it highlights the importance of understanding the cognitive deficits that may impact war survivors. Please find the concerns raised below:

1)    In the abstract, please write “dentate gyrus/hilus” and not DGH. Even though is a classic acronym, not all readers may be familiar with it.

2)    In the introduction, the authors should slightly introduce the concept of primary and secondary injury of TBI, considering that the biological evaluation was performed 30 days after the trauma.

3)    In the introduction, I understand the impact of this work in a war scenario, but considering the use of paraoxon as a pesticide, I suggest that the authors also highlight the importance of understanding the effects of a mild TBI and exposure to organophosphates in general.

4)    In the 2.1 section of materials and methods, authors state that: “all cages were moved to the experimental room for exercise and anxiety reduction”. What do the authors mean by “exercise”? Is it habituation?

5)    Please better explain the control groups. Control group (A) received vehicle injections and was anesthetized in a similar manner that mTBI animals? Did mTBI alone (B) received vehicle injections? Before or after the injury? Did the paraoxon-alone group (C) was also anesthetized in a similar manner that mTBI animals?

6)    Do the behavioral assessment was performed during the light or dark cycle? How long after the damage/injection was the EPM, NOR, and Y maze assessed?

7)    Please further explain the parameters used for the quantification of immunohistochemistry using image J (threshold for density? Analyze particles? Cell counts for positive number/mm2?). How the process length for GFAP analysis was performed?

8)    I strongly suggest that the authors display the individuals' values in the graph (with dots within the column, for example) to better visualization of intra-group variability.

9)    Please add the ANOVA statistic information throughout the results, as performed for EPM (F value, df…).

10)  Authors state that: “We found that there were significantly better performances during Y-maze testing when compared to the Paraoxon group only (p=0.018) to Control”. This sentence is hard to understand since paraoxon actually performed worst when compared to control.

11)  Was one-way ANOVA used to compare the behavioral data as well?

12)  In some of the graphs shows “paraoxon” and “paraxon” (DGH NeuN is an example).

13)  How do the authors explain a decrease in NeuN-positive cells in the paraoxon + mTBI, but not in the mTBI + paraoxon group?

14)  The discussion is a little disappointing. Many great results were not fully explored in this section. Why do the authors choose to compare paraoxon administered before and after TBI? How does the increase in astrocytes intensity relate to the decrease in the number of astrocytes in the paraoxon group? Is it related to the increase in astrocytic process length? Are there results suggestive of classically activated astrocytes (neuroinflammation, increased pro-inflammatory cytokines)? Please, further explain the neuronal loss and astrocytic data.

Author Response

Dear Reviewer,

We would like to extend our sincere gratitude for the time and effort you have dedicated to reviewing our paper. Your feedback was invaluable to us, and we appreciate your thoughtful evaluation of our work.

Your thorough and constructive analysis has aided us in enhancing the quality and precision of our research, and we are truly grateful for your meticulous attention to detail. Your expert feedback has served as a catalyst for our continued exploration of this significant topic.

Reviewer 2 Report

The manuscript by Freidin and colleagues addresses an interesting topic, given the possibility of the combined organophosphate toxicity and mild traumatic brain injury to occur in specific contexts, and the need to understand the molecular and behavioral effects to establish treatment approaches. The Authors present correlatable and complementary results from behavioral and immunohistochemical studies.

Despite its current flaws, which I detail below, I believe that this paper fits the scope of Biomedicines.

General remarks:

·      I recommend that the manuscript is revised for formatting, since some aspects have not been accurately adapted to the Microsoft Word template of Biomedicines (e.g., font type and size have inconsistencies, the way how equations/math formulas have been inserted and formatted, reference style formatting, blank sections – “Author Contributions”, “Funding”, “Institutional Review Board Statement”, “Informed Consent Statement”, “Data Availability Statement”, “Conflicts of Interest”).

·      According to the instructions for authors, acronyms/abbreviations/initialisms should be defined the first time they appear in each of three sections: the abstract; the main text; the first figure or table. When defined for the first time, the acronym/abbreviation/initialism should be added in parentheses after the written-out form. Please make sure this is done for all acronyms/abbreviations/initialisms (e.g., “mTBI” and “DGH” in the Abstract section).

Materials and Methods:

·      How were the total number of animals in the study and the number of animals per group defined? Please add some remarks on sample size.

·      Please provide a brief rationale for the choice of paraoxon dose, route of administration, duration of treatment and timepoint analyzed.

·      Why weren’t the behavioral tests and the immunohistochemistry assays performed in the same animals? This would allow an even more direct correlation between behavioral and molecular alterations. Please briefly comment on this experimental approach choice.

·      Were the behavioral and cognitive tests performed in commercially available apparatuses or were these “home-made”? In addition, how was the time spent in each compartment measured? Please add these specifications to the text.

·      Lines 209-210: please specify cellSens and ImageJ software versions.

·      Table 1: if possible, please slightly reformat the table in order to evidence the antibodies that were used together (primary + secondary antibody used to evidence the same target).

Results:

·      Figures 1, 2 and 3 are mentioned, but, probably due to an omission or formatting issue, they cannot be found within the manuscript, and there are no supplementary files. These figures must be incorporated into the manuscript.

Discussion:

·      Please suggest additional biomarkers and techniques that could be used in order to further characterize the molecular and cellular alterations deriving from mTBI/paraoxon exposure.

·      Please provide a rationale for the choice of hippocampal (DGH) and temporal cortex tissue sections (instead of those from other brain structures) to perform immunohistochemistry. This information can be added to the Introduction or Discussion section.

Author Response

(The authors gave the same response as above.)

Reviewer 3 Report

In this paper authors utilized a closed-head  traumatic brain injury model of a mouse induced by a weight drop device and an organophosphate in order to study spatial and visual memories , as well as immunohistochemistry analysis , were measured after  days from exposure  to mouse. Molecular and cellular changes were assessed within the temporal cortex and hippocampus.With very mild traumatic brain injury, using any paraoxon combination, significantly resulted in functional deficits. These results indicate additive damage caused by the combination of mTBI and paraoxon exposure.All effects were studied. The results obtained in this study are in concordance with research conducted after the Gulf War on soldiers.The results are clear and the paper is well written.

Author Response

Dear Reviewer,

We wanted to express our sincere gratitude for your thoughtful and gracious review of our paper. Your comments and feedback were incredibly valuable, and we appreciate the time and effort you put into reading and evaluating our work.

Round 2

Reviewer 1 Report

The authors have sufficiently answered all the questions raised.

Author Response

Thank you for your comments; language and a fine/minor spelling check have been conducted. 

Reviewer 2 Report

I thank the Authors for carefully addressing my concerns. However, I am still not able to see any figures within the manuscript provided. There is no additional document.

Author Response

Thank you for your comments. An additional spell check has been conducted. 

Figures are attached in a file below. Please see the attachment.  
